# Role of Micronutrient Supplementation in Promoting Cognitive Healthy Aging in Latin America: Evidence-Based Consensus Statement

**DOI:** 10.3390/nu17152545

**Published:** 2025-08-02

**Authors:** Carlos Alberto Nogueira-de-Almeida, Carlos A. Cano Gutiérrez, Luiz R. Ramos, Mónica Katz, Manuel Moreno Gonzalez, Bárbara Angel Badillo, Olga A. Gómez Santa María, Carlos A. Reyes Torres, Santiago O’Neill, Marine Garcia Reyes, Lara Mustapic

**Affiliations:** 1Medical Department, Federal University of Sao Carlos, São Carlos 13565-905, Brazil; dr.nogueira@me.com; 2Geriatrics Unit, San Ignacio University Hospital, Pontifical Javeriana University, Bogotá 110231, Colombia; ccano@javeriana.edu.co; 3Department of Preventive Medicine, Paulist School of Medicine, Federal University of São Paulo, São Paulo 04023-062, Brazil; lrramos1953@gmail.com; 4Buenos Aires and Argentine Society of Nutrition, Favaloro University, Buenos Aires C1044ABE, Argentina; 5Department of Nutrition, Diabetes and Metabolism, Faculty of Medicine, Pontifical Catholic University of Chile, Santiago 8331150, Chile; manuel.i.moreno@gmail.com; 6Facultad de Ciencias para el Cuidado de la Salud, Universidad San Sebastián, Santiago 7510157, Chile; bangel@inta.uchile.cl; 7Public Nutrition Unit, Institute of Nutrition and Food Technology, University of Chile, Santiago 7830490, Chile; 8Center for Research in Health Science (CICSA), Faculty of Health Science Anahuac University, Ciudad de México 52786, Mexico; draogb@gmail.com; 9School of Medicine, Oncology Hospital of Coahuila, Autonomous University of Coahuila, Saltillo 25280, Mexico; nutricioncarlosreyes@gmail.com; 10School of Medicine, Instituto Tecnológico y de Estudios Superiores de Monterrey (ITESM), Ciudad de México 14380, Mexico; 11Institute of Neurosciences, Favaloro Foundation University Hospital, Buenos Aires C1078AAB, Argentina; santiago.oneill@gmail.com; 12Haleon, Ciudad de Mexico 01900, Mexico; marine.x.garcia@haleon.com; 13Haleon Ltd., Victoria B1644BCD, Buenos Aires, Argentina; lara.x.mustapic@haleon.com

**Keywords:** cognitive aging, Latin America, multivitamin and mineral supplementation, Delphi method, consensus statements, healthy aging, elderly

## Abstract

**Background:** Cognitive decline is a growing public health concern in Latin America, driven by rapid aging, widespread micronutrient inadequacies, and socioeconomic disparities. Despite the recognized importance of nutrition, many older adults struggle to meet daily dietary micronutrients requirements, increasing the risk of mild cognitive impairment (MCI). This study aimed to establish expert consensus on the role of Multivitamin and Mineral supplements (MVMs) in promoting cognitive healthy aging among older adults in Latin America. **Methods:** A panel of nine experts in geriatrics, neurology, and nutrition applied a modified Delphi methodology to generate consensus statements. The panel reviewed the literature, engaged in expert discussions, and used structured voting to develop consensus statements. **Results:** Consensus was reached on 14 statements. Experts agreed that cognitive aging in Latin America is influenced by neurobiological, lifestyle, and socioeconomic factors, including widespread micronutrient inadequacies (vitamins B-complex, C, D, E, and minerals such as zinc, magnesium, chromium, copper, iron and selenium), which were identified as critical for global cognitive function and brain structures, yet commonly inadequate in the elderly. While a balanced diet remains essential, MVMs can be recommended as a complementary strategy to bridge nutritional gaps. Supporting evidence, including the COSMOS-Mind trials, demonstrate that MVM use improves memory and global cognition, and reduces cognitive aging by up to 2 years in older adults. **Conclusions:** MVMs offer a promising, accessible adjunct for cognitive healthy aging in Latin America’s elderly population, particularly where dietary challenges persist. Region-specific guidelines, public health initiatives, and targeted research are warranted to optimize outcomes and reduce health inequities.

## 1. Introduction

As per the World Health Organization (WHO), the global population aged 60 years and older is projected to double from one billion in 2020 to two billion by 2050 [1]. For the first time in history, there are now more older adults than children globally [2]. Latin America is one of the regions experiencing the most rapid aging, where individuals over 60 years currently constitute approximately 13.2% of the population, a proportion expected to reach nearly 30% by 2060 [3]. This demographic transition has intensified the focus on healthy aging, particularly in terms of cognitive health, which is critical for maintaining autonomy, functional capacity, and quality of life in older adults [4]. Cognitive aging refers to the progressive decline in functions such as memory, attention, processing speed, and executive function that occur with advancing age [5]. These changes are partially driven by neurobiological factors, such as reductions in gray matter volume, synaptic dysfunction, and diminished neuroplasticity [6]. However, they are also significantly influenced by modifiable lifestyle factors including physical activity, a healthy diet, educational attainment, and social engagement [7]. A growing body of evidence highlights the critical role of adequate nutrition, particularly the sufficient intake of essential vitamins and minerals, in preserving cognitive health and delaying age-related decline.

In Latin America, older adults often face multiple barriers to achieving optimal nutritional status. Food insecurity, limited healthcare access, and socioeconomic inequalities contribute to widespread micronutrient inadequacies, especially in B-complex vitamins, vitamin D, vitamin E, magnesium, and zinc [2]. These inadequacies have been mechanistically linked to cognitive impairment through pathways involving oxidative stress, inflammation, and homocysteine accumulation [2].

Multivitamin and mineral supplements (MVMs) offer a practical strategy to help address these nutritional gaps, particularly among older adults with increased physiological needs or inadequate dietary intake. While not a substitute for a healthy diet, MVMs are increasingly recognized as valuable complementary interventions for maintaining cognitive function. Clinical trials, including the COSMOS-Mind (Cocoa Supplement and Multivitamin Outcomes Study for the Mind) study, suggest that daily MVM use can improve global cognition and episodic memory, reducing cognitive aging by 2 years in older adults [8,9,10].

Despite these promising findings, several challenges hinder the systematic use of MVMs in the region. Dietary reference values for vitamins and minerals vary widely across countries, and comprehensive national nutrition data specific to elderly populations in Latin America remain scarce [11,12]. Moreover, healthcare providers often lack clear, evidence-based guidance on the use of MVMs in cognitive health promotion due to variations in individual health status, lifestyle, and regional contexts [13,14].

To address these critical gaps, a panel of nine experts in nutritional science, geriatrics, and neurology from five Latin America countries conducted a modified Delphi process to develop consensus statements on the role of MVMs in promoting cognitive healthy aging. This paper presents the findings of this consensus, offering evidence-based recommendations and practical insights to support public health efforts and clinical practice in Latin America.

## 2. Methods

### 2.1. Expert Panel Formation and Study Design

The panel experts were selected based on their research expertise, professional experience, and geographic representation. The final panel consisted of nine specialists in nutritional science, geriatrics, and neurology from Brazil, Argentina, Chile, Colombia, and Mexico, each with more than 10 years of experience. The panel convened virtually on the 7th, 9th, 28th, and 30th August 2024 and used a modified evidence-based Delphi process to evaluate the literature and develop consensus statements on the use of MVMs to promote cognitive health in the elderly.

### 2.2. Literature Review and Evidence Gathering

A comprehensive literature search was conducted in PubMed, MEDLINE, and Embase using a pre-specified set of key terms. The detailed search strategy and inclusion/exclusion is depicted in Appendix A. After duplicate removal and screening for relevance, 109 publications were included to support the final consensus statements (Appendix A).

### 2.3. Delphi Process and Consensus Development

The expert panel engaged in four structured virtual meetings (Figure 1 presents the timeline and process). Before every meeting, pre-reading material supporting clinical evidence was shared with the expert panel for review, ensuring all members were well-prepared and informed. This proactive approach fostered a dynamic exchange of ideas, allowing for in-depth engagement with the data and thoughtful refinement of perspectives.

Discussions were guided by a meticulous synthesis of clinical data, expert insights, and regional healthcare contexts, ensuring that the recommendations reflected both empirical evidence and real-world applicability. Through rigorous analysis and debate, the panel identified key areas of agreement and refined the proposed statements accordingly. After each meeting a structured questionnaire comprising the draft statements was shared with the experts (Figure 1).

To uphold objectivity and transparency, anonymous voting was conducted via the Qualtrics online platform. Experts rated their agreement with each statement on a 5-point scale (1 = Strongly disagree; 2 = Disagree; 3 = Neutral; 4 = Agree; 5 = Strongly agree). A consensus was defined as at least two-third agreement (≥67%).

Each finalized consensus statement was graded using the Grades of Recommendation, Assessment, Development, and Evaluation (GRADE) system to assess the strength of evidence [15]. Study designs were categorized into five levels (Level 1: high-quality RCTs and systematic reviews/meta-analyses, Level 2: prospective cohort and case–control studies, Level 3: cross-sectional studies, Level 4: case series, Level 5: expert opinion and narrative reviews) which served as the basis for evidence grading (Figure 2). Accordingly, each consensus statement was assigned a GRADE rating (A–E), reflecting its corresponding study level and the degree of agreement among experts. This structured approach, combining expert opinion with systematic evidence review, resulted in the development of 14 region-specific consensus statements for guiding MVM use to support cognitive healthy aging in the Latin America’s elderly population aged 60 years and older, representing a spectrum of educational attainment (primary through tertiary) and household incomes (low to high socioeconomic strata). They were assumed to be free of major comorbidities, such as uncontrolled cardiovascular disease, advanced diabetes complications, significant neurodegenerative disorders, or active malignancy.

## 3. Results

### 3.1. Consensus Statement Development

Using the modified Delphi methodology, the expert panel developed 14 consensus statements across four thematic domains: (A) Healthy Aging and its Relationship to Cognition, (B) Cognitive Functions and Brain Structures involved in Cognitive Healthy Aging, (C) Multivitamin and Mineral Supplementation (MVM) for Cognitive Health, and (D) MVMs as a Complementary Public Health Strategy. These statements were developed through four online meetings and multiple rounds of discussion and voting, summarizing each consensus statement, the level of agreement among experts, the supporting evidence, and the GRADE-assigned level of evidence.

#### 3.1.1. (A) Healthy Aging and Its Relationship to Cognition

This section encompasses four consensus statements supported by cross-sectional studies, randomized controlled trials (RCTs), systematic reviews, and meta-analyses to explore the prevalence, demographic determinants and risk factors, significant barriers, and technology-based interventions associated with managing cognitive health among the elderly in the Latin America region, as shown in Table 1.

**Statement 1:** Cognitive impairment has increased in Latin America in the elderly over the past few years.**Agreement:** Strongly Agree: 6, Agree: 3; Level of evidence: C

Mild cognitive impairment (MCI) has emerged as a growing public health concern across the Latin America region. A systematic review and meta-analysis involving 20,220 participants across nine countries, including Brazil, Mexico, Argentina, Colombia, Peru, Cuba, the Dominican Republic, Venezuela, and Costa Rica, reported a wide prevalence range of all-type MCI, from 6.8% to 25.5% [17]. Another meta-analysis estimated the prevalence of MCI in Mexico at 18% for men and 21% for women [18]. Similarly, a population-based study in Mexico reported that cognitive impairment prevalence increased dramatically with age from 10% among individuals aged 50–59 years to 55% in those aged 80–89 years [19].

Subtypes of MCI also vary, with amnestic MCI reported at 5.9%, MCI-dysexecutive at 4.2%, and MCI-visuospatial at 7.7% [13]. Notably, the prevalence of both MCI and dementia in Latin America is significantly higher than in high-income countries, such as those in Europe and the United States, and this trend is projected to continue [17].

According to a 2024 study, approximately 54% of dementia cases in Latin America are attributable to 12 modifiable risk factors. This is notably higher than the global estimate of 40% reported by the 2020 Lancet Commission on Dementia. The study emphasizes the urgent need for early, targeted prevention strategies to address these risk factors and reduce the burden of dementia in the region [20].

**Statement 2:** As people age, cognitive impairment generally becomes more pronounced, and typically noticeable between the ages of 60 and 70 years. However, underlying neurobiological changes may begin as early as 40 years old.**Agreement:** Strongly Agree: 4, Agree: 5; Level of evidence: A

Advancing age is a well-established risk factor for cognitive decline [6]. Epidemiological studies across Latin America consistently demonstrate a positive correlation between age and MCI incidence [14]. A systematic review by Pais et al. revealed that studies including participants aged ≥60 years exhibit significantly higher incidences of cognitive impairment. This difference was statistically significant, highlighting that the likelihood of cognitive decline increases notably with age, particularly after 65 years [21].

Emerging neuroimaging research indicates that subtle cognitive and structural brain changes may start in midlife. In a study by Kennedy et al. a notable distinction between young (20–39 years) and middle-aged adults (40–59 years) was observed in the midline superior frontal cortex, where middle-aged individuals showed increased modulation in response to cognitive demands [22]. This finding suggests the possible emergence of compensatory neural mechanisms beginning during midlife [6]. Complementary findings from Flanagan et al. highlighted that the brain undergoes neural development until approximately the age of 30, followed by a gradual onset of atrophy [23]. Although measurable neurodegeneration and related clinical symptoms often become apparent much later, these structural changes begin subtly and may accumulate over decades [23].

Cognitive aging, therefore, is a gradual process that unfolds over the lifespan, with its roots tracing back to early adulthood. While noticeable cognitive impairment, such as declines in memory and executive function, often becomes apparent around the age of 60, the damage can start at 40 years old [22]. By the time individuals reach their 60s, these accumulated changes become more pronounced, particularly in areas like problem-solving, multitasking, and long-term memory. Structural brain changes, such as reduced hippocampal volume and prefrontal cortex shrinkage, further contribute to this decline [24].

Importantly, individual variability plays a significant role, with factors like education, physical activity, mental engagement, and diet influencing the trajectory of cognitive aging. Recognizing that cognitive decline begins early underscores the importance of proactive measures, such as maintaining a healthy lifestyle including diet, to promote brain health and mitigate the impact of aging on cognitive function [25].

Drawing on clinical expertise and evidence from studies by Kennedy et al. and Flanagan et al. the expert panel emphasized that while cognitive impairment often becomes clinically apparent around age 60, the underlying pathological changes may begin decades earlier, potentially as early as age 40 [22,23].

**Statement 3:** The most significant barriers to addressing cognitive impairment in the elderly in the Latin America region are underdiagnosis, time constraints, and resource limitations.**Agreement:** Strongly Agree: 6, Agree: 3; Level of evidence: A

The Latin America region continues to face multiple challenges in basic telecommunication infrastructure, especially in rural areas [26]. Underdiagnosis is a major barrier to managing cognitive impairment in the elderly as patients often hesitate to discuss cognitive concerns, which complicates early identification. Furthermore, evaluating cognitive impairment is time-consuming, and the short duration of primary care visits makes thorough assessments challenging. Resource limitations exacerbate these challenges, including a lack of specialists like geriatric psychiatrists and neurologists, especially in rural areas. The scarcity of diagnostic tools and support services further hamper effective management [27]. The technology-based platforms play a pivotal role in sustaining cognitive functions in older adults over the long term [28]. Cognitive training apps, such as Lumosity, CogniFit, and BrainHQ, offer scientifically designed exercises to enhance memory, attention, and problem-solving skills. These engaging activities are tailored to individual needs, helping users maintain and potentially improve cognitive abilities [29]. Telehealth services further support cognitive health management by enabling older adults to consult HCPs from home. This approach is particularly beneficial for individuals with mobility challenges or those living in remote areas. Remote monitoring devices also facilitate the tracking of vital signs and health status changes, ensuring timely medical interventions [26,30]. Social isolation poses a significant threat to cognitive health and may cause dementia, but digital platforms help older adults stay connected with loved ones, reducing loneliness and depression [20,31]. Moreover, integrating digital inclusion with physical activity has been shown to enhance global cognition among elderly populations [32]. These interconnected technologies are needed to preserve cognitive health in elderly in Latin America region.

**Statement 4:** Several risk factors such as poor nutrition, unhealthy lifestyle, smoking, excess alcohol consumption, obesity, and lack of awareness contribute to cognitive decline in the elderly population.**Agreement:** Strongly Agree: 9; Level of evidence: A

Cognitive impairment in older adults is a multifactorial condition influenced by a wide range of modifiable and non-modifiable risk factors. These include poor dietary habits, advanced age, medical conditions, sedentary lifestyle, smoking, excessive alcohol consumption, obesity, mental health challenges, education, social isolation, and gender differences (Appendix A) [25,33]. Recent data from seven Latin American countries (Argentina, Brazil, Bolivia, Chile, Honduras, Mexico, and Peru) evaluated 12 modifiable risk factors for dementia such as less education, hearing loss, hypertension, obesity, smoking, depression, social isolation, physical inactivity, diabetes, excessive alcohol intake, air pollution, and traumatic brain injury. The study reported that the overall population attributable fraction (PAF) for dementia was 54%, with obesity (7%), physical inactivity (6%), and depression (5%) being the most impactful contributors [20].

National and regional studies further support these findings. In Argentina, low educational attainment and older age have been identified as key determinants of neurocognitive disorders [26]. Ramos-Henderson et al. reported that age, depression symptoms, bone fractures, widowhood, unemployment, and having a family member with dementia were significantly associated with increased cognitive impairment [34]. Additionally, González-Carballo et al. reported that women consistently exhibited higher prevalence rates of MCI across all age groups compared to men in Mexico [19]. Conversely, several protective factors were noted to mitigate the risk of cognitive decline. These include adherence to a balanced diet, higher levels of education, consistent hypertension treatment, social engagement, and regular participation in cognitively stimulating and physical activities [35].

Despite the growing body of evidence on both risk and protective factors, a significant gap in awareness and education remains. Many older adults in Latin America are unaware about the role of nutrition and lifestyle in maintaining brain health. For instance, the benefits of antioxidant-rich foods, omega-3 fatty acids, and essential vitamins in cognitive preservation are not widely known or practiced [36,37,38,39,40].

Recognizing this unmet need Latin America has launched several initiatives, including Lifestyle Intervention to Prevent Cognitive Decline (LatAm-FINGERS) which stands out as the region’s first non-pharmacological, multicenter RCT. Inspired by the successful Finnish Geriatric Intervention Study to Prevent Cognitive Impairment and Disability (FINGER), LatAm-FINGERS aims to evaluate the effectiveness of a structured, multidomain lifestyle intervention including diet, exercise, cognitive training, and cardiovascular risk monitoring in preventing cognitive decline among older adults in Latin America [41,42].

#### 3.1.2. (B) Cognitive Functions and Brain Structures Involved in Cognitive Healthy Aging

This section comprises four consensus statements, supported by cross-sectional studies and systematic reviews of RCTs. These statements focus on the Mediterranean-DASH Intervention for Neurodegenerative Delay (MIND) diet, food insecurity, and the role of vitamins and minerals in cognitive impairment, as shown in.

**Statement 5:** Vitamins and minerals are neuronutrients. Vitamin C, D, E, B-complex, chromium, copper, iron, magnesium, selenium, and zinc are micronutrients that have specific roles in brain structures and global cognitive functions.**Agreement:** Strongly Agree: 9, Level of evidence: A

Neuronutrition, a field within nutritional neuroscience, underscores the critical role of micronutrients in modulating cognitive performance and brain health. Extensive evidence from clinical and mechanistic studies confirms that specific vitamins (C, D, E, and B-complex) and minerals (chromium, copper, iron, magnesium, selenium, and zinc) directly influence neurophysiological processes such as neurotransmitter synthesis, synaptic plasticity, oxidative stress mitigation, and inflammation control.

B vitamins (B6, B9, B12) regulate homocysteine metabolism: deficiencies are linked to increased brain atrophy and cognitive impairment [43].Vitamin D modulates neurotrophic factors such as Brain-Derived Neurotrophic Factor (BDNF) and demonstrates neuroprotective properties, with low levels associated with dementia risk [44].Vitamin C supports dopamine synthesis and acts as a potent antioxidant [45].Vitamin E stabilizes neuronal membranes by preventing lipid peroxidation, implicated in Alzheimer’s pathology [46].

Among minerals:
Zinc is crucial for synaptic transmission and hippocampal function [47].Magnesium modulates N-methyl-D-aspartate (NMDA) receptors, essential for learning and memory [48].Iron supports myelination and dopamine production: its early-life deficiency causes long-term cognitive deficits [49].Selenium, through glutathione peroxidase, enhances antioxidant defenses and its deficiency correlates with cognitive decline [50].Copper is required for neurotransmitter synthesis, while its dysregulation is implicated in neurodegeneration [51].Chromium indirectly enhances neuronal glucose utilization by improving insulin sensitivity [52].

These micronutrients are not only biologically essential but clinically relevant. In the COSMOS Clinic Subcohort, multivitamin–mineral supplements (containing Vitamins C, D, E, B-complex, zinc, copper, magnesium, and selenium) significantly improved global cognition (*p* = 0.001) and episodic memory (*p* = 0.04) over two years in older adults [8,9,10]. Given their specific neurobiological functions and demonstrated impact on cognitive outcomes, these vitamins and minerals unequivocally qualify as neuronutrients. Their optimal intake supports healthy cognitive aging, while deficiencies are associated with cognitive deterioration, underscoring their necessity in aging-related nutritional strategies.

**Statement 6:** Dietary pattern from the MIND diet is effective in preventing cognitive decline.**Agreement:** Strongly Agree: 3, Agree: 6; Level of evidence: A

The MIND diet, which integrates key elements of the Mediterranean and DASH (Dietary Approaches to Stop Hypertension) diets, has been associated with a reduced risk of cognitive decline in several observational studies (Appendix A) [35,53]. This dietary pattern emphasizes nutrient-dense foods, including leafy greens, berries, nuts, whole grains, fish, and olive oil, which are rich in neuroprotective compounds such as vitamin E, folate, carotenoids (e.g., lutein), and flavonoids (e.g., anthocyanins). These nutrients help mitigate oxidative stress, inflammation, and neurodegeneration, which are key drivers of cognitive aging (Appendix A) [36,37,38,39,40]. However, despite strong observational data supporting the neuroprotective benefits of the MIND diet, RCTs establishing a causal relationship remains limited. Additionally, its specific applicability and efficacy within Latin America require more context-specific research. Differences in genetic background, traditional diets, food accessibility, and cultural practices may influence adherence and outcomes. Ongoing initiatives such as Latin America-FINGERS are expected to provide critical insights into the feasibility and effectiveness of the MIND diet in Latin American populations.

**Statement 7:** Adherence to the MIND diet is challenging in Latin America due to economic, cultural, and structural barriers.**Agreement:** Strongly Agree: 4, Agree: 5; Level of evidence: A

Adhering to the MIND diet across Latin American countries is difficult due to several inter-related barriers. Economically, key components of the MIND diet such as green leafy vegetables, berries, nuts, whole grains, olive oil, and fish are often cost-prohibitive for many households. In several Latin American countries, studies have shown that the most under-consumed MIND diet components among older adults such as olive oil and red fruits are also among the most expensive and least accessible items in the region [36,37,38,39,40].

Cultural preferences and traditional culinary practices also pose significant obstacles. The dietary patterns in many Latin American countries often diverge from the MIND diet framework, which emphasizes foods not commonly consumed or easily incorporated into traditional diets. Therefore, promoting the MIND diet in the region requires not only improving access but also ensuring cultural acceptance and relevance [38]. A study conducted among cognitively healthy older Mexican adults further supports these findings. It showed that consumption of core MIND diet foods, especially olive oil, berries, and fish, was low, primarily due to cost and limited availability [38]. Additionally, a national survey from Mexico highlighted that the consumption of sweetened beverages remains alarmingly high across all age groups, while the intake of healthy foods is comparatively low [54]. Similarly, national surveys conducted in Argentina have also shed light on a concerning trend of the frequent consumption of non-recommended, unhealthy foods, coupled with a significant 41% decrease in fruit consumption within the population [55].

To overcome these barriers it is essential to adapt the MIND diet based on local dietary habits, economic constraints, and food availability. Culturally tailored dietary recommendations, alongside targeted nutritional education and economic policies to improve food access, are critical for enhancing adherence. The MIND diet, known for its neuroprotective benefits, has shown potential in delaying cognitive decline, and its impact may be strengthened through adjunct use of MVMs, particularly in regions where nutrient inadequacies are common and may contribute to MCI [39]. Recognizing this challenge, the LatAm-FINGERS trial is crucial in evaluating the MIND diet’s role in cognitive health among Latin America’s elderly population [41].

**Statement 8:** Food insecurity in Latin America should be considered a risk factor for cognitive impairment.**Agreement:** Strongly Agree: 4, Agree: 5; Level of evidence: A

Food insecurity has been identified as a critical and modifiable risk factor for cognitive decline among older adults in the Latin America region. The prevalence of food insecurity in the region is alarmingly high, with 36.4% of the population experiencing inconsistent access to nutritious food [56]. A systematic review by De la Cruz-Gongora’s et al. identified considerable variability in micronutrient deficiencies across Latin America aging populations, noting their significant effects on cognitive performance, frailty, and bone mineral density [2]. Complementary findings from Saenz et al. reported that Mexican adults experiencing persistent food insecurity had worse cognitive performance, particularly in verbal learning, verbal recall, visual scanning, and verbal fluency [57,58].

Socioeconomic disparities including poverty, low education, and limited access to healthcare further exacerbate food insecurity in the region. The high cost of nutrient-dense foods limits the ability of many older adults to maintain a balanced diet, thereby increasing the risk of neurodegeneration and dementia. Inadequate intake of micronutrients essential for brain function contributes to structural and functional brain changes, highlighting the role of nutritional sufficiency as a protective factor for cognitive aging [59,60]. Anti-poverty programs in Mexico aimed at improving nutrition and reducing food vulnerability have shown positive impacts on food security and cognitive health [61]. These findings underscore the importance of implementing region-wide strategies to mitigate food insecurity and promote cognitive resilience in aging populations. While the current body of evidence, including cross-sectional and RCT data, supports this association, longitudinal studies are required to establish the long-term cognitive benefits of improving food security. Policymakers should prioritize nutrition-sensitive interventions to address the intertwined challenges of aging, poverty, and cognitive decline in Latin America.

#### 3.1.3. Multivitamin and Mineral Supplementation (MVM) for Cognitive Health

This section comprised three consensus statements supported by cross-sectional studies and reviews of randomized controlled trials focusing on the prevalence of micronutrient deficiencies and the role of MVMs in promoting cognitive healthy aging.

**Statement 9:** The elderly Latin American population has a high insufficiency of vitamins and minerals.**Agreement:** Strongly Agree: 4, Agree: 5; Level of Evidence: A

Multiple national population-based surveys, cross-sectional studies, and systematic reviews consistently highlight widespread insufficiency of key vitamins and minerals including Vitamin E, Vitamin D, calcium, Vitamin A, magnesium, and Vitamin C among adolescents, adults, and the elderly population in urban areas of Argentina, Brazil, Chile, Colombia, Ecuador, Peru, and Venezuela in the Latin America region (Appendix A) [59]. These insufficiencies/deficiencies are not only common but clinically significant as they compromise neuronal function, synaptic plasticity, and cognitive performance [22,43]. Addressing these widespread insufficiencies/deficiencies is crucial to support brain health and improve quality of life in aging populations.

**Statement 10:** Meeting micronutrient RDAs through diet alone is challenging for older adults.**Agreement:** Strongly Agree: 3, Agree: 6; Level of evidence: A

Aging introduces unique physiological and lifestyle barriers that make it increasingly difficult for older adults to meet recommended dietary allowances (RDAs) through food intake alone. These include reduced appetite, decrease in fruits and vegetables, and an increase in unhealthy food, dental limitations, altered taste perception, and restricted diets due to chronic conditions or medications; for this reason, it is difficult to cover the daily requirements of these nutrients. For instance, the RDA for vitamin D increases from 15 µg/day in younger adults to 20 µg/day for those aged over 70 years. Despite this, vitamin D deficiency remains highly prevalent in the elderly due to reduced sun exposure and diminished skin synthesis [62]. Recent reviews of cross-sectional studies have linked these inadequacies to accelerated cognitive decline in Latin America’s aging populations [36]. Daily MVM use offers a viable solution to bridge these dietary gaps and support cognitive health.

**Statement 11:** Daily MVM use promotes cognitive healthy aging.**Agreement:** Strongly agree: 4, Agree: 5; Level of evidence: A

Clinical evidence indicates a high prevalence of micronutrient deficiencies in the Latin America region, largely driven by insufficient dietary intake and low supplement use. These inadequacies, even when subclinical, can impair cellular and physiological functions that are critical for maintaining cognitive health. One well-established biomarker of cognitive risk is elevated plasma total homocysteine (tHcy), which is modifiable through targeted nutritional interventions [43]. Clinical studies report that the relative risk of dementia increases by 1.15 to 2.5 times in individuals with moderately elevated tHcy (within the normal range). The population-attributable risk for cognitive decline linked to raised tHcy ranges from 4.3 to 31% [63]. Intervention trials in elderly individuals with cognitive impairment show that supplementation with folic acid, B vitamins (B6, and B12), and vitamins C, E, and D reduces the total homocysteine (tHcy) level [62]. This reduction is associated with decreased rates of global and regional brain atrophy and a slowing of cognitive decline [43]. Appendix A outlines the biological mechanisms by which elevated tHcy contributes to cognitive deterioration, both directly and indirectly. Reverse causality (represented by dashed line) could also explain the association between tHcy and cognitive impairment [62]. Daily supplementation of vitamins and minerals promotes cognitive healthy aging (Appendix A). Based on this evidence, the expert panel affirms that daily MVM use, when taken within recommended levels and not exceeding the Tolerable Upper Intake Level (UL), is a safe and effective approach to meet micronutrient needs and support cognitive function in older adults [14]. MVMs are particularly valuable for delivering essential vitamins and minerals that are commonly under-consumed in the Latin American population, making them a practical strategy to promote healthy cognitive aging. The panel highlighted landmark findings from the COSMOS study, which demonstrated that daily MVM use led to statistically significant improvements in global cognition (*p* = 0.0009) and episodic memory (*p* = 0.0007) [8,9,10]. Most notably the cognitive benefits were substantial, equivalent to slowing age-related decline by approximately 2 years. These results underscore MVMs’ potential not just for basic nutrition, but also as a proactive measure for maintaining mental sharpness in aging populations.

#### 3.1.4. MVMs as a Complementary Public Health Strategy

This section encompasses three consensus statements, supported by cross-sectional studies and RCTs. A public health strategy to provide MVMs is recommended to ensure daily micronutrient requirements are met. This approach is particularly necessary to reduce MCI in the Latin America region [64].

**Statement 12:** Considering that the MIND diet pattern is difficult to cover in Latin America’s elderly population, an appropriate strategy for cognitive healthy aging is using MVMs as a preventive method.**Agreement:** Strongly Agree: 9; Level of evidence: A

Achieving the dietary patterns of the MIND diet is challenging for elderly populations in Latin America due to limited access to diverse and nutrient-rich foods available through the Mediterranean diet. As a result, many elderly individuals experience age-related cognitive decline exacerbated by inadequacies/deficiencies in essential micronutrients such as B vitamins, vitamins D and E, magnesium, and zinc that support neuronal function and synaptic plasticity [22,43].

In this context, MVM supplementation emerges as a viable strategy to bridge nutritional gaps and promote cognitive health. Robust evidence from trials like the COSMOS study demonstrated that daily MVM use in individuals aged over 60 years leads to improvements in global cognition (*p* = 0.0009) and episodic memory (*p* = 0.0007). The magnitude effect on global cognition was equivalent to reducing cognitive aging over 2 years [8,9,10].

Although MVMs are not substitutes for a healthy diet, they are widely regarded by both clinicians and consumers as an effective way to support nutritional adequacy [12]. When taken within the recommended daily allowance (RDA), they are generally safe for long-term use, with minimal risk of excessive intake even when considering dietary contributions and food fortification [14]. Moreover, public health strategies aimed at ensuring equitable access to MVMs could help mitigate micronutrient inadequacies/deficiencies in vulnerable populations like the elderly [8,9,10].

However, the successful implementation of MVM-based interventions depends on addressing socioeconomic determinants such as poverty, healthcare infrastructure, educational disparities, and geographic isolation. Targeted policies are essential to ensure that MVM use reaches those most at risk. While evidence for MVM efficacy in the general population remains mixed, individuals with pre-existing micronutrient insufficiencies are consistently shown to benefit more substantially [8,9,10].

Future research should prioritize long-term clinical outcomes, cost-effectiveness analyses, and context-specific implementation models to strengthen the evidence base and inform policy decisions across Latin America.

**Statement 13:** MVM use could be a good public health strategy for the elderly.**Agreement:** Strongly Agree: 4, Agree: 5; Level of evidence: A

MVM supplementation represents a promising public health strategy to support cognitive health in older adults, particularly those unable to meet nutrient requirements through diet alone. Evidence suggests that closing micronutrient gaps in this population may reduce the risk of cognitive decline and associated healthcare costs [64,65]. However, the effectiveness of MVMs depends on socioeconomic factors, including poverty, education levels, healthcare access, and geographic disparities. These factors influence both the affordability and equitable distribution of MVMs. Hence, targeted policy initiatives are needed to ensure that vulnerable populations benefit from supplementation programs [13,14].

Integrating MVMs into national nutrition and aging strategies offers a scalable and cost-effective solution, especially when implemented through existing healthcare infrastructures. The simplicity of a once-daily regimen supports high adherence, and long-term studies indicate a favorable safety profile [8,9,10]. Moreover, by providing a standardized nutritional safety net, MVMs can help reduce disparities in cognitive health outcomes among older adults in Latin America.

Economic modeling further supports this approach, projecting substantial savings through the delayed onset of cognitive impairment and reduced dementia-related healthcare expenditures [66]. The converging evidence from clinical trials, epidemiological studies, and economic viability strongly supports the inclusion of MVM supplementation in healthy aging policies across the region.

**Statement 14:** COSMOS-Mind studies are the most relevant ones related to MVMs and healthy cognitive aging.**Agreement:** Strongly Agree: 6, Agree: 3; Level of evidence: A

The COSMOS-Mind trials among the most comprehensive and relevant clinical investigations evaluating the role of MVMs in cognitive aging. These large-scale, long-duration studies specifically examined the impact of daily MVM and cocoa flavanol supplementation on cognitive function in older adults [8,9,10].

In the COSMOS-Clinic sub-study, a two-year longitudinal trial was conducted with 573 participants aged over 60 years, in which participants underwent in-person cognitive assessments. The results demonstrated statistically significant improvements in episodic memory (*p* = 0.0007) for those receiving multivitamin supplementation, with modest benefits in global cognition (*p* = 0.0009) [8,9,10]. These findings were further substantiated by a meta-analysis that integrated data from COSMOS-Clinic and two additional cognitive sub-studies, covering over 5000 participants. The pooled results confirmed that MVM users consistently outperformed the control groups in assessments of global cognition and episodic memory [8,9,10]. Based on this robust evidence, the expert panel concluded that MVMs play a beneficial role in supporting cognitive health in aging populations. These findings highlight the importance of incorporating MVMs as part of a broader preventive strategy against cognitive decline, particularly in regions like Latin America where the accessibility and affordability of dietary interventions are critical [8,9,10].

## 4. Discussion

This paper underscores the urgent need to address micronutrient insufficiencies among older adults in Latin America, a region marked by demographic aging, socioeconomic disparities, and limited access to nutrient-dense diets [33]. The clinical experience of the expert panel supported by a growing body of epidemiological and mechanistic evidence confirms that deficiencies in key vitamins and minerals, particularly B-complex vitamins, vitamin C, D, and E, magnesium, and zinc, are associated with accelerated cognitive decline and impaired brain function in the elderly [67,68].

As aging progresses, physiological and neurobiological changes, including reduced appetite, impaired nutrient absorption, diminished neuroplasticity, increased inflammation, and structural brain atrophy, compound the challenge of maintaining optimal micronutrient levels. These age-related alterations not only increase nutritional vulnerability but also contribute directly to cognitive deterioration [6,69]. Several Latin American population-based studies have validated these associations, further highlighting the role of micronutrient status as a modifiable determinant of healthy cognitive aging. Although a balanced, nutrient-rich diet is ideal for supporting cognitive health, its consistent adoption remains challenging in Latin America due to economic limitations, cultural dietary habits, and widespread food insecurity. In this context, MVM use offers a practical and evidence-supported solution to bridge nutritional gaps, particularly among older adults with suboptimal intakes or increased physiological needs.

Findings from high-quality trials, including the COSMOS-Mind studies, demonstrate that MVM use can yield significant cognitive benefits, such as improvements in global cognition (*p* = 0.0009) and episodic memory (*p* = 0.0007), and the magnitude of effect on global cognition was equivalent to reducing cognitive aging by 2 years [8,9,10]. These effects are especially pronounced in individuals with low baseline micronutrient status. However, it must be emphasized that MVMs are not a substitute for a healthy lifestyle; rather, they should be integrated into a comprehensive strategy that includes dietary improvements, physical activity, and cognitive engagement.

The panel also highlighted a lack of region-specific diagnostic standards and clinical guidelines. Cutoff points for cognitive screening tools like the Montreal Cognitive Assessment (MoCA) and Mini-Mental State Examination (MMSE) vary across populations and have not been adequately validated in Latin America contexts [70]. This contributes to the underdiagnosis of MCI and delays in implementing preventive strategies. The panel noted a critical need for harmonized, localized diagnostic criteria that reflect the unique sociodemographic and educational profiles of older adults in this region.

Furthermore, although some countries in Latin America have established national initiatives with micronutrient fortified foods, the panel observed that these programs largely focus on children and reproductive-age women [71,72]. The nutritional needs of the elderly are often overlooked in policy frameworks, despite clear evidence linking late-life malnutrition to cognitive decline. Current public health efforts tend to prioritize caloric sufficiency, failing to address the ‘hidden hunger’ caused by poor micronutrient quality in aging diets [13,14].

To address these gaps, the expert panel developed a set of consensus-based, evidence-informed recommendations intended to guide policy, clinical practice, and public health interventions. The panel strongly advocated for a multi-tiered approach, combining improved dietary quality, targeted MVM use, culturally appropriate nutritional education, and health system strengthening to enhance cognitive outcomes in Latin America’s elderly population. Moreover, investment in robust, large-scale randomized trials across Latin America is urgently needed to generate context-specific data that can inform practice and policy. In parallel, a recent systematic review reported that interventions combining long-chain Polyunsaturated Fatty Acids (PUFAs), specifically Eicosapentaenoic Acid (EPA) and Docosahexaenoic Acid (DHA), and selected probiotic strains (e.g., Lactobacillus and Bifidobacterium) were associated with significant improvements in global cognition, memory, and executive function in older adults. These adjunctive nutritional strategies merit further investigation alongside MVM to optimize cognitive health in aging populations [73]. Moreover, investment in robust, large-scale randomized trials across Latin America is urgently needed to generate context-specific data that can inform practice and policy.

### Strengths and Limitations

The strength of this consensus lies in its multidisciplinary and regionally diverse expert panel, which integrated clinical insight with robust evidence from both global and Latin America-specific literature. The modified Delphi methodology ensured transparency, objectivity, and regionally relevant recommendations. Furthermore, the inclusion of recent, Latin America-specific literature enhances the credibility and applicability of the findings. However, the scope of this review was limited to selected vitamins and minerals with well-established roles in cognitive function. The potential effects of other nutritional agents such as omega-3 fatty acids, probiotics, synbiotics, or coenzyme Q10 were not evaluated. Additionally, the limited number of large-scale, interventional studies in Latin America constrains the generalizability of some recommendations. Future research should prioritize regional implementation studies, cost-effectiveness evaluations, and standardized assessment protocols to further inform policy and clinical guidelines.

## 5. Conclusions

Micronutrient insufficiencies/deficiencies are prevalent among older adults in Latin America and pose a significant risk to cognitive health. This consensus highlights that essential vitamins and minerals, particularly B-complex vitamins, vitamins C, D, and E, and trace elements such as chromium, copper, iron, magnesium, selenium, and zinc, play a critical role in supporting brain structure and global cognitive functions. Despite the essential role of a healthy diet, most older adults in the region are unable to meet recommended dietary allowances through food alone due to physiological, socioeconomic, and cultural barriers. Daily MVM supplementation, when used within recommended limits, offers a practical, safe, and evidence-based strategy to help bridge nutritional gaps and support cognitive health during aging. The expert panel reached a consensus that MVM supplementation enhances memory, attention, and global cognitive performance (equivalent to reducing cognitive aging) and may serve as a valuable component of public health strategies targeting aging populations. To optimize impact, there is a pressing need for standardized, region-specific guidelines, targeted education initiatives, and continued research to evaluate long-term outcomes. Public health systems must integrate MVM interventions with broader lifestyle strategies to reduce inequities and promote healthy cognitive aging across Latin America. 

## Figures and Tables

**Figure 1 nutrients-17-02545-f001:**
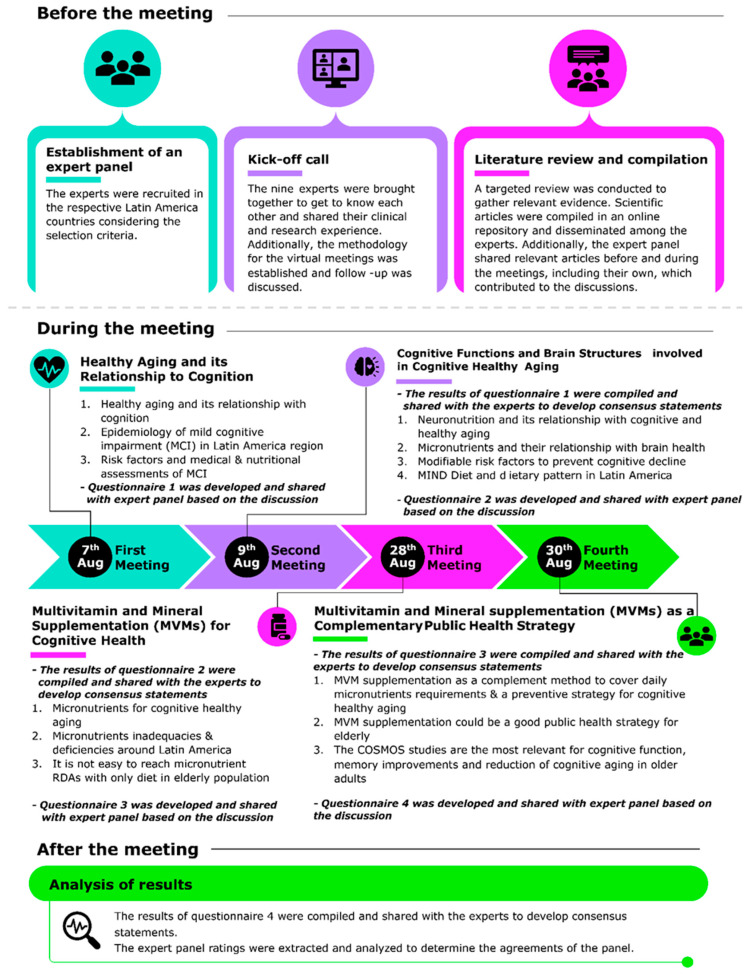
Depiction of steps involved before, during, and after the meeting.

**Figure 2 nutrients-17-02545-f002:**
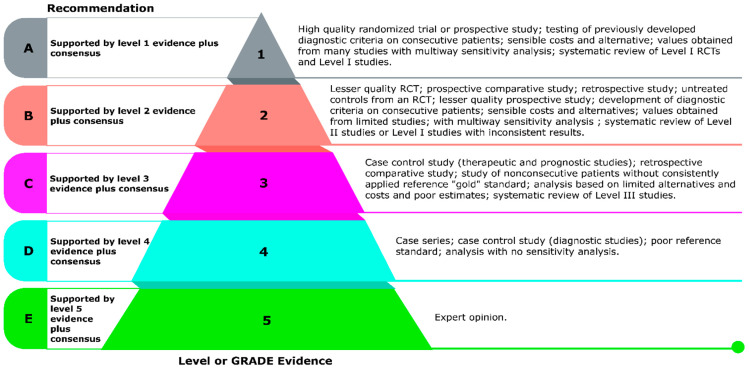
Strength of evidence using the Grades of Recommendation, Assessment, Development, and Evaluation (GRADE) system [16].

**Table 1 nutrients-17-02545-t001:** Consensus recommendations for Healthy Aging and its Relationship to Cognition in elderly in Latin America.

S. No.	Consensus	Agreement	Published Evidence	Level of Evidence
**Healthy Aging and its Relationship to Cognition in elderly in Latin America**
**1.**	Cognitive impairment has increased in Latin America in the elderly over the past few years.	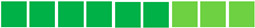 Strongly agree: 6, Agree: 3	1 SR of CSS, 4 CSSs	C
**2.**	As people age, cognitive impairment generally becomes more pronounced, and typically noticeable between the ages of 60 and 70 years. However, underlying neurobiological changes may begin as early as 40 years old.	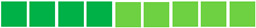 Strongly agree: 4, Agree: 5	1 SR, 1 review on RCTs, 1 clinical study	A
**3.**	The most significant barriers to addressing cognitive impairment in the elderly in the Latin America region are underdiagnosis, time constraints, and resource limitations.	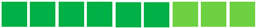 Strongly agree: 6, Agree: 3	1 SR, 6 CSSs, 1 RCT	A
**4.**	Several risk factors such as poor nutrition, unhealthy lifestyle, smoking, excess alcohol consumption, obesity, and lack of awareness contribute to cognitive decline in the elderly population.	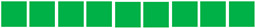 Strongly agree: 9	4 CSSs, 1 RCT	A
**Cognitive Functions and Brain Structures involved in Cognitive Healthy Aging**
**5.**	Vitamins and minerals are neuronutrients. Vitamin C, D, E, B-complex, chromium, copper, iron, magnesium, selenium, and zinc are micronutrients that have specific roles in brain structures and global cognitive functions.	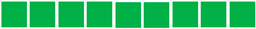 Strongly agree: 9	3 RCTs, 7 CSSs	A
**6.**	Dietary pattern from the MIND diet is effective in preventing cognitive decline.	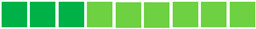 Strongly agree: 3, Agree: 6	6 CSSs, 1 SR on RCTs and CSSs	A
**7.**	Adherence to the MIND diet is challenging in Latin America due to economic, cultural, and structural barriers.	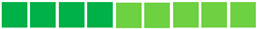 Strongly agree: 4, Agree: 5	2 CSSs	A
**8.**	Food insecurity in Latin America should be considered a risk factor for cognitive impairment.	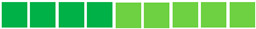 Strongly agree: 4, Agree: 5	1 SR, 3 CSSs	A
**Multivitamin and Mineral Supplementation (MVM) for Cognitive Health**
**9.**	The elderly Latin American population has a high insufficiency of vitamins and minerals.	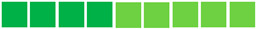 Strongly agree: 4, Agree: 5	21 CSSs, 1 review of RCTs	A
**10.**	Meeting micronutrient RDAs through diet alone is challenging for older adults.	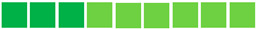 Strongly agree: 3, Agree: 6	2 reviews of RCTs and CSSs and 2 CSSs	A
**11.**	Daily MVM use promotes cognitive healthy aging.	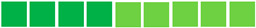 Strongly agree: 4, Agree: 5	1 review of RCTs and CSSs, 3 CSSs	A
**MVMs as a Complementary Public Health Strategy**
**12.**	Considering that the MIND diet pattern is difficult to cover in Latin America’s elderly population, an appropriate strategy for cognitive healthy aging is using MVMs as a preventive method.	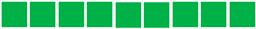 Strongly Agree: 9	2 CSSs, 2 RCTs, and 1 systematic review and meta-analysis	A
**13.**	MVM use could be a good public health strategy for the elderly.	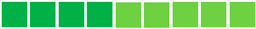 Strongly agree: 4, Agree: 5	3 RCTs	A
**14.**	COSMOS-Mind studies are the most relevant ones related to MVMs and healthy cognitive aging.	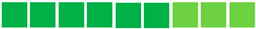 Strongly agree: 6, Agree: 3	2 RCTs and 1 meta-analysis	A

CSS: Cross-sectional study, RCT: randomized controlled trial, SR: systematic review, SR and MA: systematic review and meta-analysis, SR of CSSs: systematic review of cross-sectional studies, SR on RCTs: systematic review of randomized controlled trials, MIND: Mediterranean-DASH Intervention for Neurodegenerative Delay.

## Data Availability

Not applicable.

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
