# Peer review of "Role of Micronutrient Supplementation in Promoting Cognitive Healthy Aging in Latin America: Evidence-Based Consensus Statement"

_nutrients, 2025, doi:10.3390/nu17152545_

Round 1
Reviewer 1 Report
Comments and Suggestions for Authors
Vitamin D may be involved in immunomodulation, calcium signaling, and maintaining the blood-brain barrier. Given that the overwhelming majority of Americans and many other countries worldwide have vitamin D deficiency or insufficiency, and that vitamin D modulates neuroinflammation and potentially neurodegeneration, there is interest in whether vitamin D may improve or at least preserve cognition in aging populations. While correlational studies back this presupposition up, randomized clinical trials (RCTs) show inconsistent evidence.
The manuscript presents a white paper/consensus statement, laying out in detail recommendations for Latin America in terms of cognition at large for older adults, substrates and sequelae relevant to vitamin D and other micronutrients, whether or not multivitamin and mineral supplementation (MVMs) are useful for combatting vitamin D insufficiency or deficiency, and if MVMs could be used as a public health initiative to slow down the onset of cognitive decline or other cognitive decline related diseases associated with old age, typically Alzheimer's disease (AD) and other dementias.
The topic is highly relevant to public health and aging research. I note no major flaws.
Major strengths include:
1) a focus on a highly used over-the-counter supplement and cognitive aging, which for normal cognition or mild MCI remains relatively invariant until it is too late for clinicians to intervene with existing therapies;
2) use of only RCTs instead of associational studies;
3) Expert consensus regarding cognition at large in Latin America, related substrates and sequelae, and relationships with cognition and vitamin supplementation; and
4) subgroup analyses segmented by duration, dose, and frequency.
Author Response
Comment 1: [Vitamin D may be involved in immunomodulation, calcium signaling, and maintaining the blood-brain barrier. Given that the overwhelming majority of Americans and many other countries worldwide have vitamin D deficiency or insufficiency, and that vitamin D modulates neuroinflammation and potentially neurodegeneration, there is interest in whether vitamin D may improve or at least preserve cognition in aging populations. While correlational studies back this presupposition up, randomized clinical trials (RCTs) show inconsistent evidence.
The manuscript presents a white paper/consensus statement, laying out in detail recommendations for Latin America in terms of cognition at large for older adults, substrates and sequelae relevant to vitamin D and other micronutrients, whether or not multivitamin and mineral supplementation (MVMs) are useful for combatting vitamin D insufficiency or deficiency, and if MVMs could be used as a public health initiative to slow down the onset of cognitive decline or other cognitive decline related diseases associated with old age, typically Alzheimer's disease (AD) and other dementias.
The topic is highly relevant to public health and aging research. I note no major flaws.
Major strengths include:
1) a focus on a highly used over-the-counter supplement and cognitive aging, which for normal cognition or mild MCI remains relatively invariant until it is too late for clinicians to intervene with existing therapies;
2) use of only RCTs instead of associational studies;
3) Expert consensus regarding cognition at large in Latin America, related substrates and sequelae, and relationships with cognition and vitamin supplementation; and
4) subgroup analyses segmented by duration, dose, and frequency].
Response: We sincerely thank the reviewer for the encouraging and insightful feedback. We are grateful for the recognition of our manuscript’s relevance to public health and aging research, especially in the context of growing global interest in cognitive health strategies for older adults. The emphasis on multivitamin and mineral supplementation (MVMs), supported by randomized controlled trials (RCTs) and expert consensus, reflects our commitment to providing evidence-based, regionally relevant recommendations for Latin American. We also appreciate the reviewer’s acknowledgment of our methodological strengths, including the exclusive use of RCTs, subgroup analyses by dose and duration, and the incorporation of region-specific dietary and micronutrient profiles. These elements were central to our aim of providing clinicians, policymakers, and public health leaders with practical tools to address early cognitive decline. We thank the reviewer again for the support and encouragement.
Reviewer 2 Report
Comments and Suggestions for Authors
Thank you for the opportunity to review this manuscript. I found the topic very relevant and appreciated how the authors brought attention to an important nutritional and social issue affecting the Latin American population.
The article focuses on the possible role of multivitamin and mineral supplements (MVMs) in supporting healthy cognitive aging. The goal of the authors is to create a consensus on using MVMs as a strategy to protect brain health in older adults in Latin America.This paper addresses an important issue that highlights sociodemographic challenges.
However, this consensus is based on limited scientific evidence and applies to a population with important social and demographic differences. The use of a modified Delphi method is appropriate, but the results may not be fully evidence-based or applicable to all settings.
The article is clear, well-written, and well-organized, but I believe it could be improved by providing more details about the methodology. In particular, I would suggest the following:
- Explain how the strength of evidence was judged (e.g., RCTs vs Cross-sectional studies vs Systematic reviews and meta-analyses, reviews)
- Describe the target population more clearly (e.g. age, social background, comorbidity).
- Discuss better the role of the MIND diet, and whether MVMs are suggested for prevention or treatment of early cognitive decline (MCI).
Also, based on recent systematic reviews (Fu Q, et all, Nutrients , Oct 2024), I recommend including in the discussion the possible benefits of polyunsaturated fatty acids (PUFAs) and probiotics on cognitive health.
Author Response
Comment 1: Thank you for the opportunity to review this manuscript. I found the topic very relevant and appreciated how the authors brought attention to an important nutritional and social issue affecting the Latin American population.
The article focuses on the possible role of multivitamin and mineral supplements (MVMs) in supporting healthy cognitive aging. The goal of the authors is to create a consensus on using MVMs as a strategy to protect brain health in older adults in Latin America. This paper addresses an important issue that highlights sociodemographic challenges.
However, this consensus is based on limited scientific evidence and applies to a population with important social and demographic differences. The use of a modified Delphi method is appropriate, but the results may not be fully evidence-based or applicable to all settings.
The article is clear, well-written, and well-organized, but I believe it could be improved by providing more details about the methodology. In particular, I would suggest the following:
Explain how the strength of evidence was judged (e.g., RCTs vs Cross-sectional studies vs Systematic reviews and meta-analyses, reviews).
Response: Thank you so much for the insightful feedback.
We appreciate your recognition of the relevance and clarity of our manuscript. In response to your suggestion, we have revised the methodology section (Line no 143-148) to provide more detail on how the strength of evidence was judged. The revised text now reads:
“Study designs were categorized into five levels (Level 1: high-quality RCTs and systematic reviews/meta-analyses, Level 2: prospective cohort and case–control studies, Level 3: cross-sectional studies, Level 4: case series, Level 5: expert opinion and narrative reviews) which served as the basis for evidence grading (Figure 2). Accordingly, each consensus statement was assigned a GRADE rating (A–E), reflecting its corresponding study level and the degree of agreement among experts.”
Comment 2: Describe the target population more clearly (e.g. age, social background, comorbidity).
Response: Thank you for this valuable suggestion. We have clarified the target population in the methodology section (Line no 151-155).
“Thes population aged 60 years and older, representing a spectrum of educational attainment (primary through tertiary) and household incomes (low to high socioeconomic strata). They were assumed to be free of major comorbidities, such as uncontrolled cardiovascular disease, advanced diabetes complications, significant neurodegenerative disorders, or active malignancy.”
Comment 3: Discuss better the role of the MIND diet, and whether MVMs are suggested for prevention or treatment of early cognitive decline (MCI).
Response: Thank you so much for the suggestion. The role of the mind diet as well as the MVMs is modified now (Line no 399-402).
“The MIND diet, known for its neuroprotective benefits, has shown potential in delaying cognitive decline, and its impact may be strengthened through adjunct use of MVMs, particularly in regions where nutrient inadequacies are common and may contribute to MCI [39].
Comment 4: Also, based on recent systematic reviews (Fu Q, et all, Nutrients, Oct 2024), I recommend including in the discussion the possible benefits of polyunsaturated fatty acids (PUFAs) and probiotics on cognitive health.
Response: Thank you for this valuable recommendation.
We have updated the Discussion (Lines 628–634) to incorporate insights from the recent systematic reviews by Fu Q et al. (Nutrients, Oct 2024). The new text reads:
“In parallel, a recent systematic review (Fu Q et al., Nutrients, Oct 2024; PMID: 39458561) reported that interventions combining long‑chain PUFAs specifically EPA and DHA and selected probiotic strains (e.g., Lactobacillus and Bifidobacterium) were associated with significant improvements in global cognition, memory, and executive function in older adults. These adjunctive nutritional strategies merit further investigation alongside MVM to optimize cognitive health in aging populations.”